# Abiotic Stresses in Plants and Their Markers: A Practice View of Plant Stress Responses and Programmed Cell Death Mechanisms

**DOI:** 10.3390/plants11091100

**Published:** 2022-04-19

**Authors:** Bruno Paes de Melo, Paola de Avelar Carpinetti, Otto Teixeira Fraga, Paolo Lucas Rodrigues-Silva, Vinícius Sartori Fioresi, Luiz Fernando de Camargos, Marcia Flores da Silva Ferreira

**Affiliations:** 1Trait Development Department, LongPing HighTech, Cravinhos 14140-000, SP, Brazil; 2Genetics and Breeding Program, Universidade Federal do Espírito Santo, Alegre 29500-000, ES, Brazil; paolacarpinetti@gmail.com (P.d.A.C.); vinicius_fioresi@hotmail.com (V.S.F.); mfloressf@gmail.com (M.F.d.S.F.); 3Applied Biochemistry Program, Universidade Federal de Viçosa, Viçosa 36570-000, MG, Brazil; ottofraga@gmail.com; 4Embrapa Genetic Resources and Biotechnology, Brasília 70000-000, DF, Brazil; paololucas5@gmail.com; 5Biotechnology Department, Instituto Federal Goiano, Urutaí 75790-000, GO, Brazil; luiz.camargos@ifgoiano.edu.br

**Keywords:** plant abiotic stresses, cell markers of plant stress, plant signaling, biotechnological breeding, stress responses

## Abstract

Understanding how plants cope with stress and the intricate mechanisms thereby used to adapt and survive environmental imbalances comprise one of the most powerful tools for modern agriculture. Interdisciplinary studies suggest that knowledge in how plants perceive, transduce and respond to abiotic stresses are a meaningful way to design engineered crops since the manipulation of basic characteristics leads to physiological remodeling for plant adaption to different environments. Herein, we discussed the main pathways involved in stress-sensing, signal transduction and plant adaption, highlighting biochemical, physiological and genetic events involved in abiotic stress responses. Finally, we have proposed a list of practice markers for studying plant responses to multiple stresses, highlighting how plant molecular biology, phenotyping and genetic engineering interconnect for creating superior crops.

## 1. Introduction

Plants are continually subjected to adverse conditions that impair development and yield. As sessile organisms, they have developed sophisticated and efficient mechanisms for perceiving, avoiding, escaping and even achieving tolerance to limiting environmental conditions, typically categorized on i. biotic stress, arising from virus, fungus, bacteria and other pathogenic and/or predatory agents who parasite plants; ii. abiotic stress, regarding those adverse conditions directly associated with the fluctuation of environmental parameters such as water availability, temperature and soil composition [1,2].

Different stress responses are essentially triggered according to the severity and duration of the adverse condition. Additionally, the developmental stage also contributes tolerance or susceptibility, since some of these mechanisms are straightly associated with plant lifespan. These mechanisms integrate epigenetics, genetics, biochemical and physiological readouts, which can be temporally divided into 5 steps: i. Sensing phase; ii. Response phase—comprising the stress-sense period, typically encompassing homeostasis disruption; iii. Restitution phase—in which gene expression reprograming culminates on molecular and physiological remodeling; iv. Adaptive phase—depending on the duration and severity of stress, other mechanisms are activated, and the symptoms of plant stress are more evident; v. Regenerative phase—when stress is complete, and plant homeostasis restored [3]. Whether stress severity overcome the protective mechanisms triggered in the adaptive phase, plant cells undergo programmed cell death (PCD) and stress-induced senescence initiates [4,5].

The transition between the stress response phases, as well as during senescence, is followed by extensive gene expression reprograming. Regarding the transcription factors (TFs), they act as central players in connecting different cell signaling events and the effective changes in gene expression. Most of the products generated by the transcription of stress-responsive genes directly participate in cell homeostasis restoration, such as detoxification enzymes, besides other secondary metabolite pathways. Otherwise, early-responsive TFs regulate the expression of late-responsive ones, which regulate other genes belonging to stress responsive pathways [2].

When the environment disturbs their development, plants immediately trigger an extensive physiological remodeling coming from intricate regulatory gene networks and resulting in hormone production and release, changes in metabolites’ profile, morphological adaptations and sometimes programmed cell death. Understanding the regulatory mechanisms that govern plant responses to multiple stresses comprises one of the most important features of biotechnological agriculture, since losses associated with abiotic stresses limit plants’ geographic cultivation and severely impact agricultural yield worldwide [6,7]. Additionally, to ensure food security and agricultural sustainability, there is a growing need to turn hostile areas into productive ones [8].

Although traditional breeding strategies have provided an important contribution to agricultural crops, this approach is constrained by limited genetic diversity with restricted availability of germplasm, and this is one of the reasons why few varieties have been introduced with enhanced abiotic stress tolerance in field conditions by this methodology [9]. These advances are extremely important for rational and efficient genetic engineering approaches to improve abiotic stress tolerance in crops. Genetic modification and engineering techniques have been considered an important tool and revolutionized the development of crops tolerant to abiotic stress in recent decades [10]. The transgenic approach, based on recombinant DNA technology, includes the manipulation of regulatory genes—encoding sensors, TFs, protein kinases—which are involved in regulating stress responsive genes. Another important group used for these strategies are functional genes, which encode the osmoprotectants, detoxifying enzymes, LEA (late embryogenesis abundant) proteins and molecular chaperones, and play a direct role in mechanisms of the protective functions of plant cells [11].

Genomics and functional genomics approaches have led to a huge increase in the introduction of a variety of transgenic plants. However, transgene inactivation and improper expression have led to the failure of several transgenic plants [12,13]. The mapping of quantitative trait loci (QTLs) has been an efficient genomic technology to identify the DNA region associated with a desired characteristic. Recently, the drought resistance loci that confers germination advantage under drought stress, was identified in recombinant lines of Iranian *Oryza sativa* L. on chromosome 1 and 6 [14]. A compendium of genetically modified crops has been released by Lohani and collaborators [15], reporting performance improvement in different abiotic stresses in barley (1), rice (38), tomato (9), wheat (8), maize (5) and cotton (4) [15], encompassing different approaches, such as overexpression, co-expression or tissue-specific expression. The target genes are varied and belong to several parts of stress response pathways, including signal-sensing, transduction, gene expression regulators and effector proteins.

The most common targets arise from TFs, such as NAC, AP2/ERF, MYB and bZIPs, most of them improving drought tolerance in field conditions. Since TFs are central players in gene expression regulation and, therefore, plant physiology remodeling, it is not surprising that they are the main targets in genetically modified plant design. Other targets also include effector- and signaling- enzymes, such as peroxidases, dehydrogenases, kinases, and phosphatases, respectively. Furthermore, drought stress has been reported to induce gene expression of some metabolic pathways enzymes, such as acetyl-CoA synthetase and pyruvate decarboxylase-1. The acetyl-CoA accumulation could be used in the Krebs cycle as an energy producer or in glyoxalate cycle to produce carbohydrates. The increased expression of pyruvate decarboxylase-1 in drought-tolerant cultivars may be a consequence of the stomatal closure and the reduced photosynthetic activity during drought stress [16]. Collectively, these data reinforce the extensive crosstalk between regulatory, metabolic and developmental processes and the relevance in how the knowledge of integrative plant stress physiology may help in superior crop design.

Biotechnological agriculture has been transforming the knowledge of molecular mechanisms into improvements in cultivated crops, from the laboratory to the field. Some advances have already been made to improve the quality and production of food. In this sense, genome editing has proved to be a powerful tool, such CRISPR-Cas9, which has been successfully used in the improvement of crops such as rice, wheat, maize, tomato and soybean, to obtain specimens tolerant to abiotic stresses [17,18,19,20,21].

Gluten-intolerant consumers have created a demand for the development of low-allergen wheat. The transgene-free wheat lines engineered with CRISPR/Cas9 targeting α-gliadin genes was shown to reduce the immunoreactivity to gluten by 85% [22]. The lycopene accumulation improves tomato quality and attractive color [23]. Knowledge of the lycopene metabolism has enabled the production of lycopene-enriched tomato by CRISPR/Cas9-mediated modulation of multiple genes: increasing the lycopene biosynthesis, while inhibiting the cyclisation from lycopene to α-carotene [24]. Furthermore, genes involved in resistance to abiotic stress have been used as gene editing targets to increase food production in diverse environmental conditions in rice [25,26,27], wheat [21], and tomato [19].

CRISPR-Cas9 technology is a precision approach capable of editing a plant’s genome for introducing targeted mutation, indels and specific sequence modification using tailored nucleases. Nowadays, it has become the most important genome editing methodology for improving the quantitative and qualitative traits of the crops [28]. The availability methodologies for genome manipulation and the engineering of synthetic circuits have created new avenues for the betterment of crops, and the rational design of stress-responsive genetic circuits based on synthetic biology principles is urgently required.

The advances in understanding how plants sense and adapt to multiple stresses, as well as those players and their gene regulatory pathways, are elementary for the success of genetic engineering [6]. Accordingly, physiological, biochemical and genetic markers have been raised as hotspots for studies on plant behavior and biotechnological intervention by monitoring interesting phenotypes [29].

Hereafter, we surveyed the mechanisms involved in stress sensing and response and PCD typically evaluated in studies over plant performance under adverse conditions. Finally, we pointed out the main physiological markers and what they represent for plant stress physiology and molecular breeding in a practice view.

## 2. Plants in Adverse Environments—General Sensing and Signaling

Different abiotic stresses such as cold, heat, drought, flooding and salt can cause common cell disturbances and secondary stresses, including membrane damage, oxygen reactive species generation and damage, protein denaturation and osmotic stress at a cellular level. The versatility of the response to different unfavorable conditions in plants is related to an intricate network that comprises different levels, such as cellular, physiological and morphological defenses. Some authors suggest that the most generalized and conserved innate cellular responses in plants relies on cuticle (external protection), membrane lipid desaturation for membrane remodeling, activation of antioxidant enzymatic and non-enzymatic systems against ROS (reactive oxygen species), induction of molecular chaperones and accumulation of osmolytes compatible with cell demand [30].

These defense mechanisms are orchestrated within a complex regulatory network, involving signaling molecules, mainly the stress-related hormones (e.g., abscisic acid—ABA, ethylene—ETH), reactive oxygen species (ROS), calcium ions (Ca^2+^), hydrogen sulfide (H_2_S), nitric oxide (NO), polyamines and phytochromes. In addition, a set of effector and signaling-mediating proteins, such as the kinases, besides proteins involved in the redox balance are central players for the activation of gene regulatory transcription factors [5,31,32].

The Figure 1 introduces the general mechanisms by which plants perceive environmental stresses, transduce their signals and, throughout a sophisticated and finely coordinated response, integrate stress signals, hormonal metabolism and adaptive responses that lead to tolerance. Overall, stresses are perceived by receptors or membrane-associated proteins which trigger an ionic imbalance across the membrane, with an influx of Ca^2+^ into the cytosol. The sudden increase on Ca^2+^ concentrations in the cytosol causes a signaling cascade mediated by calcium-dependent proteins and kinases, resulting on transcription factors activation that, in turn, will modify the expression of genes related to plant physiology remodeling [33].

If the stress-combating mechanisms are not sufficiently effective in minimizing the deleterious effects of stress, mainly arising from ROS accumulation, the cells initiate environmentally-triggered cell death programs, whose global process encompasses the plant senescence [4,6,34,35,36].

### 2.1. Signaling Pathways in Abiotic Stresses in Plants

As sessile organisms, plant stress-tolerance and survival are achieved by their ability to undergo flexible and compatible responses to environmental variations. Therefore, response flexibility is governed by multilayer and timely tuned signaling pathways [37]. Many response-pathways to different stresses share the same intermediaries and metabolic branches, communicating with the primary metabolism and energy supply, which impacts plant growth and yield [36]. Even though plants can also induce unique metabolic pathways, they are typically related to metabolites produced during stressful periods, serving as readouts for studies of plant stress physiology and adaption.

### 2.2. Stress-Sensor Mechanisms in Plants

Extensive knowledge on understanding the mechanisms underlying plant stress responses has been acquired in the last decades. However, the identification of molecular sensors for abiotic stresses remains limited and many biological events might be associated with sensing mechanisms, such as changes in enzyme kinetics, membrane integrity and fluidity, cell wall architecture and molecular interactions (protein-DNA; protein-protein; ion-protein—[6,37,38].

Abiotic stresses can inflict different cell imbalances that typically converge to Ca^2+^ influx. The different ionic channels associated with calcium ion accumulation in cytosol are not necessarily directly affected by the stress condition, but also respond to secondary stimulus undergone by external imbalances. In Figure 2, we have summarized the main mechanisms converging to cytosol Ca^2+^ influx caused by osmolarity, salinity and temperature, encompassing the main abiotic stresses which plants are subjected to.

In *Arabidopsis thaliana*, the Ca^2+^ plasma membrane-associated channel OSCA1 has been primarily identified as a hyperosmolarity responsive ion channel in guard cells [39]. Its activation mechanism was recently deciphered: associated with the reduction of turgor pressure caused by water efflux triggered by hyperosmolarity, the reduced the bilateral tension of lipid bilayer opens OSCA1 ion channel, enabling Ca^2+^ transport to cytosol [40,41,42]. Downstream events are triggered by Ca^2+^ in cytosol and enhanced by ABA-dependent responses and ROS signaling, involving calcium-dependent kinases (CPKs), MAP (mitogen activated kinase proteins) and other enzymes involved in signal transduction [37,43].

Salt stress is also sensed in the plasma membrane. Microdomains enriched in phosporylcerimide can play the sensor’s role since MOCA1, a glucuronosyltransferase, adds a glucuronic acid residue at inositol phosphorylceramide, generating a negative ionic pole which can binds extracellular Na^+^ ions [44,45]. The interaction between acidic ceramides and ionic sodium at membrane induces its depolarization, opening ANN1/4 ionic channels, which allow Ca^2+^ influx. For homeostasis restoration, a downstream system headed by SOS proteins (*salt overlay sensitive*) is activated and imposes a negative feedback over ANN1/4 transporters, avoiding calcium spikes formation in cytosol [46].

Most of the Ca^2+^ flowing to the cytosol comes from the cell wall: high salinity affects the interaction between pectins and extensins. In Arabidopsis, the leucine reach repeat extensin LRX interacts with the pectins and the peptide ligand RALF, preventing its interaction with the receptor RLK (receptor like kinase)-FER. Excessive salt disrupts the interaction of RALF and LRX, disassembling the Ca^2+^-complexed pectin chains and allowing the interaction between RALF and FER, which triggers downstream signaling events [47,48].

Temperature sensing is also interconnected with Ca^2+^ as a secondary messenger. Heat affects biomolecules’ flexibility and mobility. Regarding lipid bilayers, temperature changes implicate on fluidity changes, perceived by membrane-associated proteins, such as CNGC channels (cyclic nucleotides gated channels—[49]). Additionally, temperature changes, especially heat, might lead to protein denaturation, allowing organellar proteins to eventually work as temperature sensors [37]. Misfolded proteins are recognized by heat shock proteins (HSPs) that bind heat stress responsive transcription factors (HSFs) in normal temperature. When the HSPs release from HSFs, stress responsive genes are activated and trigger downstream signals [50]. Interestingly, the promoter region of HSFs is in a special heterochromatin, whose nucleosomes enriched in H2A.Z histone-variant do not allow DNA unwrapping easily [51]. High temperatures impose H2A.Z releasing from the DNA, increasing the accessibility of transcriptional machinery to the promoter region of HSF genes and establishing a feedforward loop in controlling heat stress responses.

Differentially, cold stress is sensed by stabilization of proteins, such as the photoreceptor PhyB, which is indirectly stabilized by CBF (C-repeat/dehydration-responsive element binding factors), whose function relies on the attenuation of PIF3-mediated PhyB destruction [52]. CBF expression is dependent of the suppression of a naturally inhibitory role of ERG2 over the kinase SnRK2.6 (sucrose non-fermenting 1-related protein kinase), triggered by calcium influx through ANN1 [53,54,55].

In rice, CNCG ion channels can both sense changes in membrane properties caused by heat or cold [49]. Besides the activation of ion influx due to the imbalances in membrane fluidity, a GTP hydrolytic activity associated between COLD1 and the α subunit of RGA1, a G-protein. This interaction results in increased GTPase activity contributes to Ca^2+^ influx, composing a multilayered-controlled mechanism for cold sensing and signal transduction in response to decreasing temperatures.

### 2.3. Reactive Oxygen Species (ROS) and Their Role in Stress Signaling

The ROS encompass ions and molecules derived from atmospheric oxygen (O_2_), which, due to the instability of their unpaired electrons, display outstanding potential for reactivity with biological molecules. Although the mechanisms involving ROS in plants are not completely understood, the importance of ROS in several cellular processes is evident, including the abiotic stress responses and plant tolerance and adaptation mechanisms. The two main sources of ROS during abiotic stress are: (1) the signaling pathway, where ROS are produced for the purpose of signaling as part of the abiotic stress response signal transduction network; and (2) metabolic pathway, in which ROS are produced because of imbalances in metabolic activity. This increase in ROS can impose oxidative damage to membranes (lipid peroxidation), proteins, RNA and DNA, even leading to oxidative destruction of the cell (oxidative burst). In addition to the direct damage, ROS can generate cell toxicity caused by the formation of ROS-metabolites products that are toxic to cells [2,56].

To prevent ROS-triggered damage, plants synthetize many proteins involved in ROS detoxification, such as superoxide dismutase (SOD), catalase (CAT) and glutathione peroxidase (GPOX), as well as non-enzymatic antioxidant components, such as ascorbic acid and glutathione, which are present in almost all cells and their subcellular compartments. These components act as modulators of plant stress responses through interference between hormonal signaling, metabolic and developmental signaling. Often, genes involved in protective mechanisms of oxidative stress are used as readouts of studies of stress responses and physiology, and their ectopic expression is also applied in obtaining more tolerant plants, adapted to different conditions [56].

When acting as secondary messengers, in signal transduction, ROS are generated in response to the perception of stress, as discussed earlier. Typically, ROS production results in Ca^2+^ influx and/or phosphorylation of downstream target by stress-responsive kinases. ROS will propagate the signal throughout the changes in the redox status of different cellular components. Alterations in the structure and/or function of proteins can, for example, regulate the binding of transcription factors, and thus modulate the transcription of stress-responsive genes. Moreover, they directly change the oxidation state of cellular regulatory enzymes and their function, thus modulating different metabolic reactions [57].

Both signaling and metabolic ROS are generated in different subcellular compartments, mainly at the chloroplast, mitochondria, peroxisome and apoplast. However, each compartment establishes and controls its own homeostasis through a continuous regulation between the production and elimination of these species, generating a redox signature properly. Intriguingly, these signals are not confined to their origin: they can affect distanced ROS level, for example, by the mechanism of transport of hydrogen peroxide (H_2_O_2_) across the membrane-associated aquaporins [58]. This explains how different abiotic stresses or multiple stresses differentially impact ROS homeostasis in cell compartments, thereby generating a specific combination of signals, which determines a more adequate response to the adverse condition.

### 2.4. Ca^2+^ and Calcium-Dependent Protein Kinases (CDPKs) Cascade

Secondary messengers stand out for their ability to propagate a primary signal with great intensity and range. Among them, Ca^2+^ raises as the most important, alongside cyclic adenosine monophosphate (cAMP). Plant cells maintain cytosolic Ca^2+^ concentration between 100–350 nM, restricting the ion to ER and vacuole. Generally, in response to stimuli (biotic, abiotic, hormonal), channels in organelles and in the plasma membrane promote an influx of Ca^2+^ to the cytosol, impairing the electrochemical gradient typically maintained by the cell [51,59]. Proteins Ca^2+^-ATPases and Ca^2+^/H^+^ antiport transporters are responsible for both increasing the concentration of Ca^2+^ and re-establishing its normal level after excitation wave [35,60].

One of the most well characterized calcium sensors in plants is calmodulin (CaM). The Ca^2+^ binding to calmodulin promotes a conformational change in protein structure, activating it and allowing its interaction with several proteins involved in a plethora of essential responses, such as transcriptional and enzymatic activity regulation. Other relevant sensors belong to the group of calcineurin B-like (CBL), which are calcium-dependent serine/threonine phosphatases and interact with protein kinases to activate the Ca^2+^-dependent signaling cascade [61]. CaM and CBL are small proteins that contain multiple Ca^2+^ binding domains but lack other effectors’ domains. Hence, to transmit Ca^2+^ signal, they interact with target proteins and regulate their activity. CDPKs (calcium-dependent protein kinases) integrate a third class of proteins in calcium-mediated signaling. They act as sensors (due to their ability to perceive Ca^2+^ influx throughout Ca^2+^ binding domains) and also play an effector role, through their kinase domain [35]. In guard cells, osmotic imbalances lead to the activation of SnRK2 (sucrose non-fermenting 1-related protein kinase 2), which activates RbohD and RbohF, converging to H_2_O_2_ accumulation. From the same cascade, two specific kinases are stimulated, HPCA1, a leucine-reach repeat receptor with intrinsic kinase activity, and GHR1, resulting in Ca^2+^ spikes in cytosol [62,63]. ABA-mediated signaling and calcium activate CDPKs, which phosphorylate SLAC1, directly related to stomatal closure [64,65].

One of the most relevant results of the Ca^2+^-mediated signaling cascade in response to abiotic stresses is the activation of transcription factors, which will regulate (positively or negatively) the transcription of several genes important in the stress response. Although it is well established that the variation in intracellular concentrations of Ca^2+^ is a key event in signal transduction stimulated by stress, little is known about how different signals induce distinct and specific cellular responses.

### 2.5. SnRKs and MAPK Cascade

Reversible protein phosphorylation is one of the most important mechanisms in any signal transduction mechanism. Protein kinases play a key role in linking adaptive responses to plant stress and establishing cellular homeostasis. In plants, the family of sucrose non-fermenting 1-related protein kinases (SnRKs), divided into three subfamilies (SnRK1, SnRK2 and SnRK3), is known to participate of several response and adaptation pathways to multiple stresses, including signaling by ABA, ROS, regulation of ionic homeostasis and oxidative stress [2]. Members of the SnRK2 subfamily are serine/threonine kinases, and they are already well characterized for their roles in regulating the plant response to ABA by directly phosphorylating several downstream targets, e.g., transcription factors necessary for the expression of stress responsive genes [66].

Among the important members of the SnRK3 subfamily (also known as PKSs or CIPKs) is the salt overly sensitive 2 (SOS2) kinase, a central component of the SOS (salt overly sensitive pathway), the first abiotic stress signaling pathway established in plants. CIPKs are kinases characterized by interacting with calcineurin B-like calcium binding proteins (CBL). In the SOS pathway, which regulates the sensitivity to salinity in *A. thaliana*, SOS2, a CIPK, is activated by SOS3, a CBL, when an influx of Ca^2+^ occurs in the face of stress. The SOS3-SOS2 kinase complex acts on the intracellular homeostasis of Na^+^ and K^+^, an important mechanism for salinity tolerance. Furthermore, other studies have already demonstrated the participation of CBL-CIPK complexes in various abiotic stress signaling pathways, particularly when calcium acts as a second messenger [6,66]. In the signaling triggered by osmotic stress, we have another example of the participation of members of this family in the response and adaptation pathways to abiotic stress. The protein open stomata 1 (OST1) kinase, a member of the SnRK2 subfamily responsive to ABA that modulates the production of H_2_O_2_ via NADPH oxidase, thus controlling stomatal opening and closing [2].

Another group of kinases with a central role in signal transduction pathways in plants are the MAPKs, belonging to signaling cascades present in all eukaryotic cells. The role of MAPKs in plant immunity is already established for a several stresses, and they are better characterized in response to drought, salinity, cold, ROS and stress by heavy metals, such as cadmium and copper, besides being responsible for physiological processes of growth and development mediated by hormones [35].

The module or MAPK-cascade is usually made up of a series of three protein kinases: MAPK, MAPKK and MAPKKK. The classical activation mechanism starts with a specific signal or cell alteration that leads to MAPKKK activation. MAPKKK then works by phosphorylating MAPKK and thus activating it. MAPKK in turn phosphorylates MAPK. While active, it can translocate within the cell and phosphorylate other signal transducers, such as other kinases, metabolic regulatory enzymes or TFs, thereby regulating cell function. The initiation of signaling, with the activation of MAPKKK, can occur mediated by phosphorylation via an upstream kinase or binding with G-protein subunits, and its signal is attenuated or repressed by phosphatases [60].

Specific upstream signals can activate multiple cascades of MAPKs and at the same time these protein kinases are responsive to a wide range of signals, indicating a converging and diverging transduction network at the same time. This convergence pattern is well described when we analyze the MEKK1-MKK2-MPK4/MPK6 module activated by salinity or cold conditions, being later responsible for phosphorylation and activation of TFs, which will culminate in the expression of hundreds of genes involved in the stress response, metabolism, signaling and transcriptional regulation [67]. In guard cells, MAPK cascade overlaps the function of SnRKs, since both MKK5-MPK3/6 and SnRK2 activates SLAC1 for stomatal closure during salt induced stress [68,69].

## 3. Integrative Hormone, ROS and Kinases Signaling axis and the Signal Transduction in Abiotic Stresses

After the primary response to abiotic stresses, the releasing of secondary messengers and the remodeling of gene expression in plants prepare them for a long-distance and systemic response, leading to the plant acclimation. These findings are possible since plants produces phytohormones, which are capable of stimulating physiological adaptation as long as they are produced, besides ROS and Ca^2+^, which also participate in systemic signal transduction [37].

Convergent responses result in MAPKs activation, followed by a massive phosphorylation of several targets, mainly transcription factors, which are going to activate or repress the expression of specific stress-responsive genes. In this section, we further discuss the effects of hormones- and ROS-combined responses over physiological and transcriptional plant reprogramming in the main abiotic issues, such as drought, temperature-based, and salt stresses.

One of the main phytohormones involved in abiotic stress responses is abscisic acid (ABA). ABA is perceived by PYR/PYL receptors (pyrabactin resistance protein/PYR-like) that suppress the phosphatase activity of PP2C. Consonantly, SnRK2 activity is fully detected, positively regulating ABA-responsible targets [43], mainly the TFs involved in gene reprogramming during stress adaption or those involved in senescence-triggering [4,70,71,72]. Finally, ABA responses converge to ROS production: especially in drought, SnRK2/OST1 complexes phosphorylate Rbohs, leading to ROS accumulation [71]. Transcription factors both responsive to ABA and ROS, such as those from AP2/ERF family, are activated in response to combined hormone-ROS signals and induce downstream acclimation responses [73].

In cold, a similar mechanism is undergone in response to low temperatures: the activation of SnRK2/OST1 complex results in the transcriptional modulation and instability of ICE1 that downregulates the expression of CBF/COR genes [74,75,76], typically related to improvement on plant antioxidant system, osmolytes accumulation and plant longevity [5,77,78,79].

In heat stress, the integrated ROS-ABA axis is also activated, and downstream events have been frequently associated with thermotolerance in Arabidopsis. *ERF74*, for example, responds to ABA accumulation in heat stress and positively regulated RbohD [80]. Similarly, MBF1c, DREB2A and other AP2/ERFs TFs have shown to be differentially expressed in heat stress and participate in plant acclimation [43,81]. Interestingly, cold results on jasmonic acid (JA) accumulation, which represses the ABA-signaling. The releasing of ICE TFs activates CBF/DREB TFs, leading to a better antioxidant balance in cells and converging to a cold tolerance phenotype.

Other phytohormones also participate in plant responses to multiple stresses. Brassinosteroids (BRs) are perceived by leucine-reach repeat receptor-like kinase BRI1, which forms a heterodimer with BAK1, initiating a phosphorylation cascade that activates BRs-responsive transcription factors, such as BZR1 and BES1. BRs and ABA pathways display a discrete overlap, since BIN2 (brassinosteroid insensitive 2) kinase is activated and phosphorylates SnRK2 during drought, enhancing the expression of ABA-responsive genes [43,82,83]. Not surprisingly, ROS play a central role in connecting stress sensing and signaling, since H_2_O_2_ is also generated by BR-activated Rboh and activates BZR1 and BES1 [84,85]. Finally, the interaction between BAK1 and SnRK2 also promotes stomatal closure, demonstrating the partial convergence between BRs and ABA responses for activating drought tolerance mechanisms [86,87,88]. ROS signaling also serves as a convergence point for salicylic acid (SA) signaling in abiotic stress responses [89,90,91,92]. SA accumulation is typically associated with ROS increasing, since SA is a phytohormone that signalizes plant infections by bacteria or biotrophic fungi.

The process of precocious senescence triggered by abiotic stresses is designated ePCD (environmental programmed cell death—[93]). The role of ethylene (ETH) in senescence has been extensively described and considering the molecular bases of PCD in plants, which involve hallmarked activity of nucleases, proteases and ROS-mediated molecules’ disassembly, the role of ETH in abiotic stress responses becomes clear. EHT stimulates ROS production by activating NADPH oxidases RbohD and RbohF [94] and, thereby, overlaps other hormone-responsive pathways. In cold, ETH mediates the expression of CBF genes by EIN3, a TF classically related to senescence promoting responses [95].

The main branch of systemic signaling in plants facing stressful conditions relies on the production and releasing of hormones. ABA, SA, JA and ETH are major players in stress response and, as was extensively discussed, ABA encompasses a critical role in responses leading to osmotic and REDOX imbalances. Opposingly, SA, JA and ETH typically control biotic stress responses, but also display some important roles in coordinating stress responses and programmed cell death [5]. Precedents in the literature have reported that ABA and GA signaling pathways interact with DELLAs (acronym: aspartic acid–glutamic acid–leucine–leucine–alanine), serving as a crosstalk point [96].

The signaling pathways of SA and JA are known to intersect at various points because SA and JA regulate biotic stress responses antagonistically [97]. In contrast to the antagonistic functions of SA and JA, JA and ET operate synergistically. They positively regulate defense related genes in pathogenesis. JA and ET pathways converge to EIN3, which is also activated by abiotic stresses and integrates PCD programs [5,98].

A positive JA-ETH crossover causes induction of genes encoding proteinase inhibitors in response to wounding [99]. Likewise, JA and ETH are required to simultaneously activate the expression of ERF1, typically involved in cell cycle control, and, thereby, PR genes [100], connecting stress and developmental signal pathways. Intriguingly, ETH is also known to crosstalk with ABA in abiotic stress responses. DREBs (dehydration responsive elements binding proteins) belong to the ERF TFs’ family, genuinely induced by ethylene [101].

## 4. Secondary Metabolites as Signaling and Effector Molecules in Plant Abiotic Stresses

Secondary metabolites are organic compounds constitutively or specifically synthesized in response to different environmental conditions. They do not have a specific role in plant growth and development, but significantly affect plant tolerance to stresses [102,103]. The production and accumulation of secondary metabolites vary from plant species, environment and season, since the same plants growing in different conditions display different levels and metabolites’ composition [104,105]. For producing secondary metabolites, plants undergo stressful situations, passing by an extensive physiological remodeling to deviate compounds from the primary metabolism to specific modification that become them secondary metabolites. These processes encompass cascades of subsequent enzymatic reactions and, naturally, are energy expensive [106].

Different stresses interplay regulatory networks that converge to the same group of metabolites. Interestingly, each different group of secondary metabolites comes from a main branch of primary metabolic pathways. For example, terpenes and derived compounds arise from the mevalonic acid pathway, but can also be derived from glycolysis subproducts, such as glyceraldehyde-3-phosphate and pyruvate [103]. Differentially, phenolic compounds are synthesized using products from shikimic acid and malonic acid, besides the turnover of aromatic amino acids [107,108]. Intriguingly, more than 100,000 secondary metabolites are produced by plants subjected to environmental challenges, and this plethora of organic compounds display many biological functions, such as osmoprotectants, thermoprotectants, signaling molecules, co-enzymes, antioxidants, bio repellents and others [105,106,107].

Regarding the main abiotic stresses, salinity and drought typically converge to ionic and osmotic imbalances that lead to membrane disruption, ROS accumulation and cell-toxicity [109]. In this context, different plant systems have their content of many different secondary metabolites enriched. For example, the primary response to osmotic stress is the imbalance in sugar-starch interconversion, resulting in the accumulation of soluble sugars and polyols [110,111].

Soluble sugars and polyols can act stabilizing the cell osmotic potential serving as osmoprotectants and maintaining redox homeostasis [112,113]. Regarding sugar signaling, the enzyme hexokinase 1, which catalyzes the first step of glycolysis, acts as a glucose concentration sensor and regulates the expression of genes involved in sugar homeostasis maintenance [15]. Most of the glucose that keeps hexokinase’s activity comes from invertases activity that were reported to be linked with stress responses, mitochondrial and ROS equilibrium [113]. Trehalose, another monosaccharide, also regulates several processes related to osmotic imbalances: in small exogenous doses, trehalose reduces abnormalities caused by water deprivation exposure and upregulates the expression of several stress responsive genes, coordinating an ABA-dependent signaling pathway throughout SnRK1 [114,115,116].

Changes in amino acids metabolism have been also reported as one the strategies for plant acclimation coping with abiotic stresses. The most well studied responses involve proline and phenyl alanine amino acids in plants subjected to different stresses. Proline is water soluble and has been found to accumulate in cytosol in response to hyperosmolarity, suggesting it can pay an essential role in drought and salinity responses [117,118].

From other amino acids, plants produce a diversity of secondary metabolites involved in stress responses. Flavonoids, for example, have been associated with ROS scavenging and antioxidant activities in plants [119,120]. In rice and tobacco, treatment with exogenous flavonoids reduced oxidative damage and improved salt and drought tolerance [121,122]. Additionally, anthocyanins (derivative compounds of flavonoids pathway) also enhance stress tolerance by decreasing ROS generation [123,124].The polymerization of phenolic alcohols in lignin was suggested to be associated with drought responses, since cell wall thickening helps on cell turgor maintenance even in dehydration conditions [15]. Finally, other compounds such as rosmarinic acid, chlorogenic acid, polyamines and glycine betaines have also been associated with drought, salt and heavy metal protection, mainly enhancing the non-oxidative cell system during stress [103].

## 5. Mechanisms of Stress-Induced Senescence

Senescence is the final stage of plant development that is responsible to nutrient recycling and reallocation to the reproductive organs. In addition to being activated in developmental processes, plants can induce accelerated senescence to cope with abiotic stress. Stress-induced senescence shares some similarities in molecular mechanisms with the abiotic stress response such as (i) chloroplast degradation (ii) reactive oxygen species (ROS) signaling (iii) phytohormonal balance and (iv) transcription factors modulation.

Chloroplast breakdown is one of the most defining hallmarks of stress-induced senescence. The chloroplasts degradation affects photosynthesis and ammonia assimilation, but provides remobilization of cellular building blocks during senescence, while the nucleus and mitochondria remain intact until the final stages [125]. Abiotic stress including drought, cold, heat and salt stress can induce accelerated leaf senescence through chloroplast degradation. The STAYGREEN (SGR) and NONYELLOWING 1 (NYE1) genes are chloroplast metabolism components that regulates abiotic stress-induced senescence [126,127]. The chloroplast vesiculation (CV), a vesicular trafficking pathway that mediated chloroplast content degradation, accelerates senescence in CV overexpressed plants and increase tolerance to abiotic stress in CV-silenced mutants [128]. The cysteine proteases are crucial to regulate chloroplast protein content and their principal target is RuBisCO [129]. The cysteine protease activity was shown to be increased in leaf senescence and drought stress [130,131].

The reactive oxygen species (ROS) can act as a defense molecule at low concentrations and can induce senescence and cell death at high concentrations [132]. Abiotic stress increase ROS production that leads to higher levels of cysteine proteases activity, which in turn triggers RuBisCO degradation. Therefore, leaf senescence and abiotic stresses share ROS-mediated chloroplast degradation [133]. Furthermore, antioxidant enzymes that control ROS production and scavenging play crucial roles in tolerance to abiotic stress-induced senescence [132]. Recently, the ROS synthesis and SA modulation have been shown to be connected by a positive regulator of senescence WRKY42 transcription factor [134].

Antagonistic or synergistic phytohormonal signaling pathways can coordinate the plant response to different environments. The abscisic acid (ABA), ethylene and cytokinins (CK) can modulate the expression of genes associated with abiotic stress-induced leaf senescence and chloroplast degradation. The phytohormone ABA mediates the ROS accumulation in plants under drought stress to induce leaf senescence [135].

Under normal conditions, the ethylene levels are present in low concentrations, but during senescence and abiotic stress there are two peaks of increase in ethylene production. In the early stages, the first peak is thought to trigger synthesis of plant defensive proteins. Sometime later, the second peak is responsible to induce senescence, chlorosis and abscission [136]. The phytohormone cytokinin was shown to delay senescence and promote drought tolerance in plants [137].

In response to abiotic stress-induced senescence, the chloroplast degradation, ROS signaling and phythormone balance activates signaling cascades that culminate in transcription factors modulation. Under these conditions, transcription factors families such as NAC, WRKY, MYB, bZIP and MYB can dictate changes in the plant transcriptome to restore cell homeostasis or induce cell death [138]. The senescence-associated NAC genes (NAC-SAGs) upregulated in senescence was shown to be responsive to multiple stresses in plants [4,5]. Accordingly, overexpression of ANAC042 (JUB1) increases tolerance to many abiotic stresses and delays senescence. In addition to trigger signaling cascades in response to abiotic stress, the transcription factors WRKY can also participate in carbohydrate synthesis, senescence, development, and secondary metabolites synthesis [95]. Recently, the Arabidopsis WRKY71 was shown to regulate ethylene-mediated leaf senescence by directly activating EIN2, ORE1 and ACS2 genes [139].

Multiple stresses induce programmed cell death in plants, a situation that may persist on field conditions and underscore the relevance of cell death components as a target for engineering stress tolerance to improve plant productivity. First described in soybean, the NRP-mediated cell death signaling is a conserved pathway induced by multiple stress and senescence [140,141,142,143]. Accordingly, overexpression of the NRP-mediated cell death components (like GmNAC081, a NAC transcription factor) increase soybean susceptibility to drought stress, ER stress and senescence [144]. The overexpression of the ER resident chaperone BiP, a negative regulator of the NRP-mediated cell death signaling, enhances tolerance to drought and attenuates senescence [145]. Recently, the NRP-mediated cell death signaling was also associated with cadmium-mediated toxicity, an environment that limits plant growth [146]. Furthermore, the senescence-associated NAC genes in soybean (designates *GmNAC-SAGs*) have been mapped as potential targets to increase crop tolerance to senescence and multiple stresses [4,5]. These evidences make clear the importance of understanding the signaling pathways to obtain cultivars that are better adapted to diverse conditions.

## 6. Biochemical and Physiological Markers of Abiotic Stresses—A Practice View of Plant Phenotyping and Molecular Analyses

Constant climate changes associated with the quickly growing population create a new context for modern agriculture, demanding precise methods for evaluating plant stress responses. Regardless, molecular biology has worked as a valuable tool, allowing the identification of genes associated with relevant traits and the prospection of interesting characteristics among different plants. Stress-response pathways are intricate and sophisticated, in addition to often showing multi-layer connections with each other and between plant developmental signaling pathways. This makes the selection of markers for stress assessment a difficult task and the interpretation of results often requires extensive knowledge about the physiology and biochemistry of stress in plants. Furthermore, not all responses’ mechanisms are conserved and/or completely deciphered in all plant species, which means that there are relatively few universal markers of abiotic stress in plants.

Advances in sequencing methodologies and the continuous development and improvement of sequence manipulation techniques, integrated with biometrics methods, allowed a significant advance in plant breeding, otherwise achieved by classical breeding approaches, such as wide genome-assisted selection and biparental or multiparental crossing programs. However, these techniques might be better applied in biotechnological agricultures with the development of improved phenotyping methods [147]. The phenotype rises from the interaction between the plant and the environment and represents, practically, the set of molecular mechanisms triggered and coordinated by plant cells, reflecting on their physiology and, on a wide level, their phenotypical plasticity [148].

The main challenge of phenomics relies on the need of integrative studies for deep mechanistic understanding in the interaction between plants and local conditions, with high spatial and temporal resolution [147], allowing scientists to comprehend basic processes, determining plant performance, and creating strategies for crop management.

Phenotypical assessment can be performed surveying the plant as all (called non-invasive phenotyping) or processing harvested samples (invasive phenotyping) [147]. The plant breeding process is one of the major applications of phenotyping in practice [149]. Associated with the modernization of imaging processes, system informatization and biotechnology, the novel phenomics approaches could provide advantages in modern plant breeding, mainly regard to the sample analysis throughput, applicability under field conditions, crop-specificity, and massive trait inheritance analysis. These opportunities were achieved in the last two decades in novel sensors, automation and quantitative data analysis [150].

The transposable knowledge from model plants (reached by recombinant DNA exploiting and, more recently, gene editing tools) is currently applied in the phenotype assessment of a diversity of plants, including crops and forests species, across scales from cellular to tissue and organ level, from single plants in a greenhouse to fields. To be useful, phenotyping methods may come from extensive observations of plant behavior differentially challenged, setting standard operation procedures of experiments and extraction of data from multimodal images, generating meaningful data [147]. Further advance fructifies from the interaction between groups of plant scientists, geneticists, bioinformaticians, software engineers, data managers and breeders, reinforcing the interdisciplinarity needed in plant stress studies.

In this section, the main cellular, biochemical and gene markers for the study of adaptive responses were compiled in Table 1, as well as their analysis in various plant systems aiming to guide the complex study of physiological remodeling pathways and the analysis of data from this goal.

## 7. Final Remarks

Plant mechanisms of response to abiotic stresses are ordinarily studied in model and agronomic relevant species, revealing integrative and partially conserved responses. Even though significant advances were achieved in the last decades over these pathways in several plant systems, this field demands intensive research and exploration looking forward to understanding the complex and fascinating mechanisms of survival, adaptation and evolution of plants in the environment interaction (mainly in crops), since the use of these information in modern agriculture is not compatible with their availability and integration. Few cultivars adapted to different socio-geographic demands have been released. Several mechanisms of response and adaption in plants are conserved, but few differences between species are ordinarily identified. Hence, studies that validate data obtained in model plants and elucidate signaling and stress responsive pathways in crops may be conducted for advance of biotechnological breeding. Aiming to offer an integrative and practice view of plant stress physiology, our review highlights the main mechanisms by which plants face adverse conditions and compile them in a practice approach, offering an overview of technics and targets for studying plant stress responses in model and crop systems. 

Finally, we pointed out the necessity of multidisciplinary studies, mainly those that correlate conserved pathways already described in model plants and the phenotypes observed in crops subjected to close-related conditions for developing agronomic relevant traits. Additionally, an intensive formation with extensive practice view and accumulated experience demanded from the researchers in this advanced field, whose robust analyses are supported by knowledge on general outcomes from the community in the field, may greatly contributing for studies in crops, perennial plants, trees and fruits, for which exist an unexplored cell world in stressful conditions.

## Figures and Tables

**Figure 1 plants-11-01100-f001:**
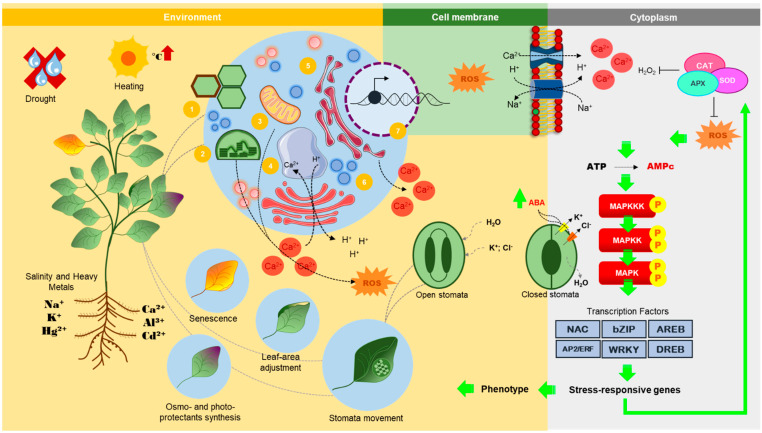
**Abiotic stresses signal integration and adaptive responses in plants.** Plants perceive different stress signals through primary receptors, which may be embedded in their plasma membrane. Primary signals also arise from the cell wall (1). Activation of these sensors often culminates in a flux of ions across the membrane that normally culminates in an influx of Ca^2+^ into the cytosol. As a localized effect of abiotic stresses, the disturbance of the energy balance of photosystems in chloroplasts (2) and of the electron transport chain in mitochondria (3) culminate in the production of ROS, which is also a response shared by other compartments, such as saturation of protein folding pathway in the endoplasmic reticulum (4) and activation of peroxidases and oxidases in peroxisomes (5). The cytosolic content of Ca^2+^ is balanced after the influx triggered by the signaling of abiotic stresses, through vacuolar antiport transporters (6). Collectively, these signals are integrated and potentiated by other secondary messengers, such as cyclic AMP (adenosine monophosphate) produced by the activation of adenylate cyclase in response to ROS and Ca^2+^ and converge to the activation of protein kinases, including calcium-dependent protein kinases (CDPKs) and cascades of MAP-kinases, which will activate hormone biosynthesis pathways, such as abscisic acid (ABA). Finally, stress-responsive transcription factors, reinforced by hormonal signaling, are responsible for the transcriptional modulation of several genes in the nucleus (7), resulting in the remodeling of plant physiology and adaptive responses that lead to characteristic phenotypes of plants under stress.

**Figure 2 plants-11-01100-f002:**
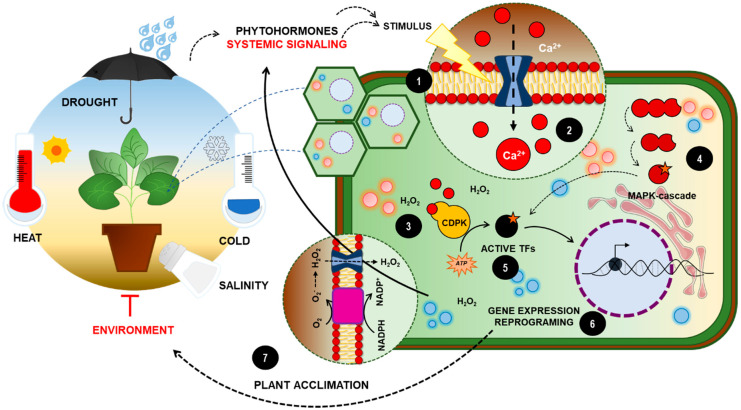
**General sensing mechanisms for abiotic stresses.** Most of the mechanisms for sensing and triggering adaptive responses to multiple abiotic stresses are associated with changes in proteins and lipids in biological membranes (1). Adverse conditions impose ultrastructural changes in biomolecules, which are sensed by receptors or specialized proteins, converging, typically, to Ca^2+^ accumulation in cytosol (2), as well as REDOX imbalances (3). These signals activate kinase cascades and other secondary events that stimulate phosphorylation/dephosphorylation cascades (4), culminating on TFs activation (5) and gene expression remodeling (6). Several enzymes involved in the biosynthesis of osmolytes, pigments, thermoprotectants and other secondary metabolites are also activated, in addition to ROS detoxifying enzymes, which attempt to reestablish redox homeostasis in cells subjected to stress (7). At the systemic level, morphological changes are detected in the root and leaves, such as the proliferation of lateral roots in response to drought, and biochemical changes, such as the secretion of phytochelants in response to heavy metals, as well as stomatal reclusion and closure, in addition to reduction of leaf area and abscission. If acclimation response mechanisms are not efficient enough to restore plant homeostasis, cells trigger early senescence, redirecting nutrient flow to reproductive tissues and seed generation through their stress-triggered cell death programs.

**Table 1 plants-11-01100-t001:** Cellular, molecular, physiological and gene markers in the adaptive response of plants to abiotic stresses.

Cell Markers			
Marker	Biological Function	Variations	Analysis
Stomatal closure	The stomatal movement, characterized by opening and closing, is a process regulated by the hormone ABA as a function of temperature fluctuation and/or water availability. The turgidity of guard cells is controlled by potassium (K^+^) and chloride (Cl^−^) channels and, according to the osmotic flow of water, the stomata may be open (high turbidity—greater osmotic potential inside the cells) or closed (low turbidity—greater osmotic potential outside the cells). The stomatal closure decreases the photosynthetic rate and transpiration. In consequence of the lower electronic flow through the photosynthetic apparatus, the generation of ROS decreases. (*)	Open/Closed	Evaluation of random fields on the abaxial surface of leaves by optical microscopy or confocal microscopy with staining of sections with propidium iodide and excitation at 543 nm [88,94,135]. (*) In cases of high light stress, ROS might to be generated, even with closed stomata, as a side effect of high electronic flow in photosystems and the imbalance in energy absorption and dissipation.
Leaf area adjustment	Leaf area adjustment is a process that, like stomatal closure, decreases the photosynthetic rate. The adjustment can be made by curling the ends or by leaf abscission.	-	Phenotypic evaluation. The profile of the leaves of a control plant (wild specimen or untreated specimen) must be carefully observed in relation to the treated specimens. To provide a quantitative parameter, it is possible to count the leaves with symptoms in relation to the asymptomatic leaves [151].
Cell wall thickening	Stress-mediated ROS accumulation leads to cell wall thickening as a way to ensure greater mechanical rigidity. This thickening is achieved through the deposition and polymerization of phenolic compounds into lignin and by modifying the structure of polymers that make up the cell wall, such as hemicellulose.	Esterification between glycoproteins and phenolic compounds	Immunolocalization mediated by specific antibodies against glycoproteins and cell wall polymers and their esterified and non-esterified variations and analysis by confocal and/or electron microscopy [152,153].
		Lignification	Quantification of phenolic compounds by Follin reaction. Mapping of genes involved in the cell wall polymer biosynthesis process [152,153].
Modifications of the lipid profile of the plasma membrane	Changes in the lipid profile of the plasma membrane in plant cells are mainly caused by temperature fluctuations and ROS-mediated lipid peroxidation.	Cold: increased concentration of unsaturated fatty acids	In practice, it is difficult to monitor changes in the lipid composition of membranes. The evaluation of these modifications can be done through indirect inference, evaluating the expression of genes related to fatty acid saturation or desaturation. HPLC (high-performance liquid chromatography) might to be an alternative with correct standardization.
		Heat: increase in the concentration of saturated fatty acids	
		Increased concentration of malonic aldehyde (MDA)	Spectrophotometric quantification at 532 nm [49,53,80].
Programmed cell death	When stress-response mechanisms fail to overcome it, plants trigger stress-induced senescence as a way of remobilizing their nutrients to reproductive organs and seeds in an attempt to perpetuate the species. The extent of senescence symptoms (chlorosis and leaf necrosis) is indicative of greater tolerance or susceptibility to stress.	Increased leaf chlorosis	Leaf chlorosis can be directly evaluated as a phenotypic parameter, observing the leaf area in which there is loss of chlorophyll or by direct spectrophotometric quantification of the pigment [134].
		Increased leaf and root necrosis	Leaf and root necrosis can be evaluated by testing the color of leaves and roots by vital dyes. In leaves, Trypan Blue or Evans Blue is normally used and the extent of cell death is directly associated with the intensity of blue staining, since the dye is only able to penetrate dead cells. In the root, this evaluation can be done by staining the roots with propidium iodide and its evaluation under confocal microscopy. In dead cells, cell wall and nucleus are stained and in living cells, only the wall is stained. Therefore, the higher incidence of stained nuclei in random fields indicates a greater extent of cell death in the root [49,153].
Programmed cell death (continued)	(continuation)	Increased degradation of proteins and nucleic acids	Degradation of proteins and nucleic acids can be assessed by quantitative and qualitative methods. The electrophoresis of nucleic acids and the total protein extract of plants subjected to stress allow us to infer on the quality of the sample, which directly reflects on the rate of degradation. Large drag areas indicate a greater degree of degradation. In the case of proteins, the rate of protein decay can be estimated by quantitative methods, such as Bradford, comparing the control sample with the treated sample, assuming as absence of degradation the amount of protein in the control sample and calculating (relatively) the rate of degradation based on the quantification of proteins in the treated sample [130].
**Molecular Markers—Hormonal Metabolism**			
**Marker**	**Biological Function**	**Variations**	**Analysis**
Abscisic acid (ABA)	ABA is the main hormone in the integration of environmental signals and adaptive physiology, controlling the expression of important transcription factors in ABA-dependent signaling pathways and stomatal closure.	Increase in ABA concentration	Spectrophotometric quantification based on immunodiagnostic kits (ELISA) or ultra-performance liquid chromatography (UPLC) [86,134,138].
**Molecular Markers—Oxidative Metabolism**			
**Marker**	**Biological Function**	**Variations**	**Analysis**
Reactive Oxygen Species (ROS)	ROS are products of metabolic processes (balanced or unbalanced), such as photosynthesis and cellular respiration, or products of the activity of peroxidases and oxidases in response to stresses. They can occur in chloroplasts, mitochondria, endoplasmic reticulum (ER) and peroxisomes.	Increase as a consequence of photosystem overload and decrease in stomatal conductance	Hydrogen peroxide (H2O2) is the most commonly analyzed form of ROS and reflects the global picture of the redox state of cells. In the qualitative evaluation, H2O2 reacts with diaminobenzamidine (DAB) or nitro-blue tetrazolium chloride (NBT) forming a brown and blue precipitate, respectively, in the leaves. Quantitatively, peroxide in acidic medium reacts with potassium iodide and is degraded to O2 and H2O. This degradation leads to a decrease in the absorbance of the reaction at 390 nm and its quantification can be performed based on a standard curve [86,134,135].
Enzyme activity (CAT, SOD, APX and GPX)	In response to ROS accumulation, plants increase the transcription and activity of antioxidant enzymes, such as catalase (CAT), superoxide dismutase (SOD), ascorbate peroxidase (APX) and glutathione-peroxidase (GPX). SOD converts superoxide radicals to H_2_O_2_ in chloroplasts. In turn, CAT and APX enzymes convert hydroxyl radicals to H_2_O and O_2_ in both chloroplasts and cytosol, as well as GPX.	Increase in enzyme activity	Enzyme activity assays based on enzyme-promoted ROS degradation and its effect on the enzymatic reaction [53,86].
Nitrogen osmolytes	Nitrogen osmolytes, mostly represented by amino acids or derivatives, play an important role in the response to abiotic stresses, providing tolerance to water loss and ROS accumulation, in addition to acting as antifreeze in cases of extreme low temperatures.	Increased concentration of proline, glycine, glycine-betaine and γ-aminobutyric acid (GABA), in addition to polyamines	Proline quantification is the most widely used technique for the quantification of nitrogenous osmolytes. It is based on the reaction of proline in sulfosalicylic acid with ninhydrin, forming a bluish colored chromophore, whose reading is taken at 520 nm and the concentration calculated based on the absorbance of the samples and the molar extinction coefficient of proline. Fractional analyzes of amino acids and other nitrogen derivatives are conducted by HPLC [53,81].
Soluble sugars	Soluble sugars, thanks to their reducing properties and their high solvation layer, are osmolytes synthesized by plants in response to drought and osmotic stress. When in high concentrations, they are able to optimize cellular osmotic potential, improve hydration status and protect lipid membranes against excessive heat and freezing.	Increased concentration of maltose, raffinose and trehalose	Detection based on oxide-reduction reactions. For the general detection of soluble sugars, the dinitrosalicylic acid (DNS) method is used, in which the reducing sugars react with the DNS when hot, oxidizing and forming a brownish-colored compound. Quantification is done by spectrophotometric reading at 540 nm and based on a monosaccharide standard curve. For quantification of specific sugars, high performance liquid chromatography is normally applied [126,130,134].
Pigments—chlorophyll	Chlorophyll comprises a group of photosynthetic porphyrin pigments present in chloroplasts that impart a greenish color to leaves. Plants have chlorophyll a and b, which differ from each other by a methyl group instead of aldehyde at position 3 of the tetrapyrrole ring.	Decrease in chlorophyll ester in stress	Direct spectrophotometric analysis using extinction coefficients properly [130].
Pigments—Carotenoids	Carotenoids are pigments of a lipidic nature, derived from terpenes. Due to their conjugated nature, they are excellent antioxidants and their abundance is associated with greater tolerance to the deleterious effects of ROS accumulation.	Decrease in carotenoid content due to oxidation	As well as chlorophyll, they are analyzed by direct spectrophotometry [130]. HPLC may also be indicated with correct standardization.
Pigments—anthocyanins	Anthocyanins are purplish pigments derived from flavonoids with a thermoprotective and antioxidant function. They are produced in response to excessive light and water deprivation.	Increased anthocyanin content in response to heat and drought	Extraction in hydrochloric acid alcohol and quantification by spectrophotometry with readings at 529 nm and 650 nm [48,80].
**Physiological Markers**			
**Marker**	**Biological Function**	**Variations**	**Analysis**
Stomatal conductance	The stomata are the main leaf structure involved in the control of gas exchange between the plant and the environment. Resistance and stomatal conductance are direct measures of the efficiency of these exchanges and reinforce the values analyzed for photosynthetic rate and transpiration, which should be lower in plants subjected to water and osmotic stress, for example.	Decrease in stomatal conductance to minimize water loss through transpiration	Infrared gas analyzer (IRGA) [154].
CO_2_ fixation Photosynthesis + Transpiration	The consumption of CO_2_ has a direct relationship with the photosynthetic rate of plants, which is reduced in abiotic stress due to the reduction in leaf area and stomatal closing and reclusion mechanisms, as a side effect of the reduction in water loss through transpiration and optimization of the use of water inside the cells.	Decrease in photosynthetic rate and transpiration rate ^1^	IRGA [154].
(1) A decrease in the photosynthetic rate at a lower rate, compared to a decrease in the transpiration rate, and associated with a decrease in stomatal conductance, indicates the existence of adaptive mechanisms that optimize the use of water and reduce its loss, characteristics often associated with better performance of plants under water stress situation.			
Relative water content	Relative water content refers to the water content of cells compared at two different points. It is generally indicative of the efficiency of water use in studies of osmotic stress, such as drought and salt stress, since higher relative water content indicates less transpiration, with a consequent improvement in the oxidative performance of cells and less leaf wilt.	Decrease in relative water content	The determination of the relative water content is done in a comparative situation, usually in irrigated plants and later submitted to water stress. The plant has its fresh mass determined and is subjected to drying to determine the dry mass and, consequently, the relative water content. [53,135]. The analysis can be also performed considering the weight of turgid leaves after water immersion in a standardized time.
Ionic flux due to electrolyte leakage	When a plant is subjected to a type of abiotic stress, normally, a flux of ions sets in across the plasma membrane in response to the stress. However, with the accumulation of ROS and the deleterious effects of stresses on the plasma membrane, there may be electrolyte leakage caused by membrane rupture. Therefore, electrolyte leakage is a parameter that infers membrane integrity.	Increased electrolyte leakage under stress	Comparative measurement in control plants and stressed plants or over time performed with a conductivity meter [95,126,134].
**Gene Markers**			
**Gene name**	**Protein/Marker Type**	**Variation**	**Analysis**
*CAT*	Catalase/Antioxidant enzyme	Increases	Gene expression analysis by qRT-PCR and expression calculation by 2-ΔCt or 2-ddCt comparative method [80,82,85,95,126,154].
*SOD*	Superoxide dismutase/Antioxidant enzyme	Increases	
*APX*	Ascorbate Peroxidase/Antioxidant enzyme	Increases	
*GPX*	Glutathione Peroxidase/Antioxidant enzyme	Increases	
*AREB-1*	ABA responsive element-binding/Transcription factor	Increases	
*DREB1/2A*	Dehydration responsive element-binding/Transcription factor	Increases	
*RD29A*	Responsive to desiccation 29A/Transcription factor	Increases	
*RD29B*	Responsive to desiccation 29B/Transcription factor	Increases	
*RD20*	Responsive to desiccation 20/Transcription factor	Increases	
*RAB18*	Ras-related protein 18/LEA protein	Increases	Gene expression analysis by qRT-PCR and expression calculation by 2-ΔCt or 2-ddCt comparative method.
*PAL*	Phenylalanine ammonia lyase/Phenylpropanoid pathway enzyme and phenolic compound synthesis	Increases	
*NRP-1/2*	N-rich protein 1 and 2/Transcription factor	Increases	
*VPE*	Vacuolar processing enzyme/Caspase-like enzyme in cell death processes	Increases	
*CNX*	Calnexin/Calcium-dependent protein	Increases	
*GLK1*	Golden like-1 protein/Chloroplast maintenance	Decreases	
*NYC1*	Non-yellow Coloring 1/Chlorophyll degradation	Increases	
*PaO*	Pheophorbide Oxidase/Chlorophyll degradation	Increases	
*BFN1*	Bifunctional nuclease 1/Ac. nucleic and proteins	Increases	
*SINA1*	Seven in Absentia/Protease	Increases	

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
