# Peer review of "Abiotic Stresses in Plants and Their Markers: A Practice View of Plant Stress Responses and Programmed Cell Death Mechanisms"

_plants, 2022, doi:10.3390/plants11091100_

Round 1

Reviewer 1 Report

The manuscript has improved. The authors integrated the discussion of some aspects related to water stress in plants.

However, I would like some aspects associated with drought to be discussed.

In particular:

  • Authors should at least cite the contribution of Drought Resistance Loci in some species (see Noryan, M. et al., Drought Resistance Loci in Recombinant Lines of Iranian Oryza sativa L. in Germination Stage. BioTech 2021, 10, 26. https: / /doi.org/10.3390/biotech10040026). I believe that identifying quantitative trait loci (QTL) in the selection of drought tolerant species is important.
  • Furthermore, drought induces some plants to regulate the gene expression of the synthesis enzymes of some metabolites. This is crucial in the process of selecting cultivars, or species adapted to water stress (see Ashrafi, M. et al .. Physiological and Molecular Aspects of Two Thymus Species Differently Sensitive to Drought Stress. BioTech 2022, 11, 8. https: / /doi.org/10.3390/biotech11020008).

In my opinion, after the brief discussion of these recent papers, the manuscript can be published.

Author Response

Dear reviewer,

We sincerely appreciate your comments and accepted your suggestion for briefly discuss about the papers you have listed.

Please, check the lines 78 – 102 and the references [14] and [16].

Reviewer 2 Report

I think that the paper represents an interesting review in the field of abiotic stresses in plants and their physiological markers. 

I suggest to accept the paper in the present form.

Author Response

Dear reviewer,

We sincerely appreciate your comments and your precise evaluation of our manuscript.

Reviewer 3 Report

There is some improvement in the manuscript quality, but it still requires some minor improvements.

L26 "avoiding and escaping" how is it differentiated according to authors, I suggest to add "tolerance."

Fig. 1. AMP is not explained.

L163-166 I do not think that such introduction is needed in one of subsections.

Fig. 2. As in fig 1 there should be direct links between description and figure using for example numbers as in Fig 1.

L218 - needs correction

L220-221 temperature changes do not necessary lead to denaturation, it needs correction

Table 1. "stomatal closure" the link between electron flow (not electronic) is not as simple, it depends on light conditions. At high light stomatal closure may induce ROS due to imbalance in energy absorption and dissipation.

Relative water content - I assume authors describe leaf relative water content. Typical procedures differs from what is written and includes also turgid weight after immersing leaf sample in water for specific time.

Author Response

Dear reviewer,

We sincerely appreciate your comments and your precise evaluation of our manuscript.

Following your suggestions for better acceptability of our manuscript, we have performed the following modifications:

L26 "avoiding and escaping" how is it differentiated according to authors, I suggest to add "tolerance."

  • Added – please, check the new sentence (in red) in line 26.

Fig. 1. AMP is not explained.

  • Added – please, check the line 175.

L163-166 I do not think that such introduction is needed in one of subsections.

  • Sentence removed.

Fig. 2. As in fig 1 there should be direct links between description and figure using for example numbers as in Fig 1.

  • Please, check the new figure attached and its caption (lines 203 – 216)

L220-221 temperature changes do not necessary lead to denaturation, it needs correction

  • Corrected – please, check the lines 243 – 245.

Table 1. "stomatal closure" the link between electron flow (not electronic) is not as simple, it depends on light conditions. At high light stomatal closure may induce ROS due to imbalance in energy absorption and dissipation.

  • For better inform the readers about this particular case, we have added a footnote. Please, check the Table 1 (information in red).

Relative water content - I assume authors describe leaf relative water content. Typical procedures differs from what is written and includes also turgid weight after immersing leaf sample in water for specific time.

  • We have also added the possibility for doing the RWC analysis using immersed samples. Please, check the table 1 (information in red).

Reviewer 4 Report

The manuscript entitled: “Abiotic stresses in plants and their markers: a practice view of
plant stress responses and programmed cell death mechanisms” is comprehensive, well written, concise and transparent discussion of the most important factors related to abiotic stress in the plant. The figures 1 and 2 are a great introduction to and explanation of the physiological and molecular mechanisms of plant stress responses.

Minor remarks:

Introduction:

Sentence started in line 45 : “ regarding, Tf…..” is too long and complicated

More precise description of examples of using technologies of transgenic approach (line 67), functional genes (line  70) transgene inactivation (line 75), as well as  genome edition in relation to improving stress resistant will be interesting here. The examples of CRISP/Cas  in food quality (gluten, gliadin etc)  was interestingly described due to giving particular examples, however without relation to topic of the manuscript.

Section 3 about secondary messengers

Please add also few sentences regarding the plant hormones cross-communication in stress response with examples, it is very interesting and  topical issue.

Section 4 about secondary metabolites

New added section about secondary metabolites in stress significantly enriched the manuscript. First sentence in not true, there are also constitutive metabolites.

Section 5 Added paragraph about multiple stress is also very topical,

here you can also mention about plant-microbes  interaction in improving stress tolerance in plants which now is intensively studied topic (examples of interesting articles: https://doi.org/10.1094/MPMI-08-20-0233-R, https://doi.org/10.1111/1365-2435.13499)

Author Response

Dear reviewer,

We sincerely appreciate your comments and your precise evaluation of our manuscript.

For better acceptability of our manuscript, we have performed the following modifications:

Sentence started in line 45 : “ regarding, Tf…..” is too long and complicated

  • Sentence was rewritten. Please, check the new sentences in lines 46 – 52.

More precise description of examples of using technologies of transgenic approach (line 67), functional genes (line  70) transgene inactivation (line 75), as well as  genome edition in relation to improving stress resistant will be interesting here.

  • Thinking on how to maintain the focus of the review in stress markers, we have surveyed some additional information on how QTLs and OGMs design can improve crop performance. We opted to do not discuss it deeply because the review we cited (Lohani et al., 20222) is a recent, complete and focused compendium of genes involved in crop obtention.
  • Please, check the lines 78 – 102.

The examples of CRISP/Cas  in food quality (gluten, gliadin etc)  was interestingly described due to giving particular examples, however without relation to topic of the manuscript.

  • Aiming to better adapt the manuscript scope, we have decided for removing these sentences.

Section 3 about secondary messengers

Please add also few sentences regarding the plant hormones cross-communication in stress response with examples, it is very interesting and  topical issue.

  • Added, Please, check the section 3 – lines 451 – 470.

Section 4 about secondary metabolites

  • New added section about secondary metabolites in stress significantly enriched the manuscript. First sentence in not true, there are also constitutive metabolites.

  • Please, check the line 472 – 473.

Section 5 Added paragraph about multiple stress is also very topical,

here you can also mention about plant-microbes  interaction in improving stress tolerance in plants which now is intensively studied topic (examples of interesting articles: https://doi.org/10.1094/MPMI-08-20-0233-R, https://doi.org/10.1111/1365-2435.13499)mersed samples.

  • Since our manuscripts runs over just about abiotic stresses, we decided to do not add more information about abiotic stresses or plant-pathogens interactions.

This manuscript is a resubmission of an earlier submission. The following is a list of the peer review reports and author responses from that submission.

Round 1

Reviewer 1 Report

The authors propose an in-depth analysis on the different responses of plants to abiotic stresses. In particular, they report the main environmental, molecular and hormonal signals that trigger the defensive responses of plants.

The authors enrich the review with interesting images and tables. The analysis is very thorough and the whole discussion is linear. However, I have not found anything innovative and above all a specific problem that has not yet been resolved. What is the added value of this review? What are the molecular mechanisms yet to be defined? What are the results achieved to date? Unfortunately these are questions that are not answered in this review. In my opinion, the manuscript may be a valid chapter in a plant physiology book, but it cannot be considered valid as a review. Authors should reorganize the manuscript, highlighting the critical issues still existing today on the subject and better discuss recent literature.

Author Response

Dear reviewer,

We sincerelly appreciate your comments over our manuscript. For improving and adequating it to the higher standards of Plants and to make it more attractive, we have deeply reviwed and added some sections, following the addressed suggestions: In the abstract, we pointed out the necessity of multidisciplary studies and transposable knowledge in plant stress reponses for modern crop design, better presented in the Introduction (please, check 72 - 114 lines). Additionally, we have improved the manuscript with a section of Secondary Metabolites and their importance in plant adaption (please, check the section 4) and the challenges in phenotyping applied to plant stress diagnosis (please, check the section 6). Finally, we have addressed some questions of biotechnological breeding in our final remarks (in red).

With this modifications, we hope to transform the manuscript into a valuable review in plant stress physiology.

Reviewer 2 Report

The review paper by de Melo et al describes molecular processes underlying the plant response to abiotic stress. The topic of the review is important, because plants as sessile organisms should constantly combat the limiting environmental condition (abiotic and biotic stresses).

The manuscript is well structured. The review’s structure reflect consequent stages of stress response in plants: response, restitution, adaptive and regenerative phases. This structure helps reader to follow the involvement of different molecular mechanisms in response to stress.

Authors made good illustrative material (figs. 1 & 2) that summarize in a graphical form main molecular players in plants involved in abiotic stress response. This help to understand more clearly the review.

The part that is most attractive from my point of view, however, it the section 10 describing systematically markers of stress. It contains very informative table 1 which summarize in a clear manner Cellular, molecular, physiological and gene markers in the adaptive response of plants to abiotic stresses and how the symptoms of plant stress response could be observed.

The review includes 90 literature references, almost half of them (42) published in last 5 years.

I think that this work worth publishing in Plants, however, I have some recommendations that may improve the manuscript before publishing.

Major:

(1) The section 10 and Table 1 are very good contribution by the authors. However, my recommendation is to extent this section by brief description of how modern plant phenotyping methods helps to evaluate the stress severity and plant response to abiotic stress. This looks reasonable, because the column ‘Analysis’ in the Table 1 represents, in general, the phenotyping methods to detect physiological changes in plants.

(2) I recommend to supplement Table 1 by references to the papers in which the methods of stress marker detection and usage was successfully demonstrated or to the protocol papers that describe the implementation of stress markers detection. This will increase the value of the table 1 data. This may not be done, however, for all the table rows: for example, I don’t think that each gene marker need a single reference. But reasonable additional information would be beneficial.

Minor:

(3) Please provide list of abbreviations

(4) Line 180: it is better to change “vibratory frequency” to “thermal fluctuations”

(5) I recommend checking carefully the manuscript for typesetting errors, seeing for example:

Line 173: “cytosol comes from de (the?) cell wall

Line 219: “plants undergo a large number of proteins” ,  the meaning is unclear

Line 456: Table caption formatting error

Author Response

Dear reviewer,

We sincerely appreciate your comments for amending our manuscript.

Follow the modifications addressed, according to your suggestions:

Major:

  • The section 10 and Table 1 are very good contribution by the authors. However, my recommendation is to extent this section by brief description of how modern plant phenotyping methods helps to evaluate the stress severity and plant response to abiotic stress. This looks reasonable, because the column ‘Analysis’ in the Table 1 represents, in general, the phenotyping methods to detect physiological changes in plants.

Following your suggestion, we have added some paragraphs introducing how modern phenotyping techniques may help the understanding on how plants respond to different abiotic stresses. Please, check the section 6.

  • I recommend to supplement Table 1 by references to the papers in which the methods of stress marker detection and usage was successfully demonstrated or to the protocol papers that describe the implementation of stress markers detection. This will increase the value of the table 1 data. This may not be done, however, for all the table rows: for example, I don’t think that each gene marker need a single reference. But reasonable additional information would be beneficial.

Table 1 was completely referenced.

Minor:

  • Please provide list of abbreviations

Following the structure provided in journals’ guidelines, we did not add an abbreviations list, but we carefully revised the manuscript to include their meaning alongside the text.

  • Line 180: it is better to change “vibratory frequency” to “thermal fluctuations”

To avoid equivocated use of scientific concepts, we have reformulated the sentence. Please, check the lines 220 – 221.

  • I recommend checking carefully the manuscript for typesetting errors.

We have revised the manuscript and eventual misspellings were corrected.

Reviewer 3 Report

The Authors of the manuscript aimed to surveyed mechanisms involved in plant stress sensing and response to abiotic stresses. It is not easy task due to complexity of processes and their interrelations. There is a need for such review as there is constant improvement in many, often distant areas of plant science. But in my opinion the current version of the review needs serious improvement.

The problem is that some of the topics were described in detail and some of them mitted or described too briefly. Due to this it is sometimes hard to follow the text.

Some detailed most important issues:

  1. The abstract is very short some additional information on the novelty of approach is needed.
  2. L25 - I suggest to skip any information on abiotic information, the scope given in tittle is clear
  3. L29 plant response to yield is not always possible I would rewrite the sentence
  4. L41. I would suggest to add "sensing phase" - it is in the review
  5. L 66 in the range of described abiotic stresses, light stress is missing. I would suggest to add it as it has many common with other stresses, but this is not necessary only a suggestion.
  6. Fig 1. I have the impression that picture is not consistent with caption - it should be rewritten. Part of the caption should be in manuscript body.
  7. L125 the numbering of teh topic should be changed. Topic no.3 looks like one of main topics whereas 4 a subtopic.
  8. Fig. 2 It is difficult to understand the figure and the cascade of linked processes. Also the links with description are not clear. The mechanisms not described shouldn't be shown. MAPK is in chapter 7. Abbreviations are not explained
  9. L175 Here and in other places, abbreviations are not described it makes difficult to follow.
  10. L193 on the other hand?
  11. L194 in the context of the proteins, adding description of the light stress would be justified https://doi.org/10.1016/j.tplants.2020.06.010
  12. earlier?
  13. L311, L320 structure of many paragraphs should be improved as the review should not only collect information from different sources but also  compare them. In teh manuscript many paragraphs describe study and finish it with single citation.
  14. Table 1 - needs strong improvement. I think it should not contain detailed description of physiological processes - it should be in the manuscript. turbidity - turgidity; stomatal closure do not results in decrease in ROS, at least not always; some methods are missing or are not well described - it should be indicated that examples of methods are given, or more methods should be included. methods should be given with references. The analysis of chlorophyll is not clear to me. Why in carotenoids HPLC method is not indicated? Relative water content should be completely rewritten, the method described is not commonly used.
  15. L 471-476 arguments not really convincing. Based on the review I would point out sth. related to the need of interdisciplinary studies

Author Response

Dear reviewer,

We sincerely appreciate your comments for amending our manuscript.

Follow the modifications addressed, according to your suggestions:

  • The abstract is very short some additional information on the novelty of approach is needed.

We have added some information in abstract, mainly those from the necessity of multidisciplinary studies and the role of modern biotechnology into obtaining superior crops, highlighting how the knowledge over plant stress physiology may help it.

  • I suggest to skip any information on abiotic information, the scope given in tittle is clear

We have reformulated the introduction, omitting some information about abiotic stresses and adding others over how biotechnology and molecular biology, combined to the breeding, can enhance the releasing of adapted crops. Please, check the section 1.

  • plant response to yield is not always possible I would rewrite the sentence

Sentence rewritten – please, check the line 62.

  • I would suggest to add "sensing phase" - it is in the review

Added – please, check the line 45

  • I have the impression that picture is not consistent with caption - it should be rewritten. Part of the caption should be in manuscript body.

Figure caption was rewritten.

  • the numbering of teh topic should be changed. Topic no.3 looks like one of main topics whereas 4 a subtopic.

The first division in sections and subsections was automatically done by the journal formatting. Please, check the new sections in the manuscript.

  • 2 It is difficult to understand the figure and the cascade of linked processes. Also the links with description are not clear. The mechanisms not described shouldn't be shown. MAPK is in chapter 7.

The figure 2 represents a general connection between sensing and downstream responses in plants subjected to stressful conditions. To offer a better understanding of these connection to the reader, we have reformulated the figure caption. Please, check.

  • Abbreviations are not explained

We carefully revised the manuscript and abbreviations were explained alongside.

  • L193 - on the other hand?

The sentence was rewritten – please, check the line 234.

  • structure of many paragraphs should be improved as the review should not only collect information from different sources but also  compare them. In the manuscript many paragraphs describe study and finish it with single citation.

Most of the paragraphs were revised, the references adequate and more recent information added.

  • Table 1 - needs strong improvement. I think it should not contain detailed description of physiological processes - it should be in the manuscript.

The idea of a short description of the biological process in table 1 concerns on facilitate the reader to correlate the analysis, the phenotype and the process evaluated without reading the whole manuscript. It makes the table more useful and practice as a guide for plant stress response analyses.

  • turbidity - turgidity;

Corrected.

  • stomatal closure do not results in decrease in ROS, at least not always;

Rewritten

  • some methods are missing or are not well described - it should be indicated that examples of methods are given, or more methods should be included. methods should be given with references.

The Table 1 was completely referenced.

  • The analysis of chlorophyll is not clear to me.

The formatting of the table by journals’ system had suppressed this information. It was corrected.  

  • Why in carotenoids HPLC method is not indicated?

Added as an alternative method in carotenoids and other analysis, when applicable.

  • Relative water content should be completely rewritten, the method described is not commonly used.

Rewritten.

  • L 476 - arguments not really convincing. Based on the review I would point out sth. related to the need of interdisciplinary studies

We have reformulated the final remarks, addressing your suggestion. Please, check the lines 663 – 666 and 670 – 673.

Reviewer 4 Report

The manuscript entitled: ’Abiotic stresses in plants and their markers: a practice view of plant adaption and programmed cell death mechanisms’’ is comprehensive, well written, concise and transparent discussion of the most important factors related to abiotic stress in the plant. The figures 1 and 2 are a great introduction to and explanation of the physiological and molecular mechanisms of plant stress responses. Nevertheless, some basic information is missing. First of all, there is lack of references and examples on how this knowledge about molecular mechanism of plant response to stress  have been used to improve the resistance in example:

Line 32 ‘’Understanding the regulatory mechanisms that govern plant responses to multiple stresses comprises one of the most important features of biotechnological agriculture…’’’, here some example of practice usage will greatly enrich the manuscript.

On the other hand, the plant breeding programmes and improving resistance to abiotic stresses by this should not be omitted here as well.

Differences between plant adaptation and acclimatization should be defined, explained, highlights and both terms should be used in proper meaning. Adaptation is mentioned in title, but there is no reference to adaptation in manuscript, please add it.

Differences in molecular mechanism  between highly resistant and stress-sensitive varieties or cultivars should be described in order to highlight the importance of described molecular and physiological markers.

There is a lack of secondary metabolism description as one of the important element of molecular  response of plants to stress.

Table 1 should be signed and properly formatted, appropriate references  of usage should be added.

Author Response

Dear reviewer,

We sincerely appreciate your comments for amending our manuscript.

Follow the modifications addressed, according to your suggestions:

  • First of all, there is lack of references and examples on how this knowledge about molecular mechanism of plant response to stress have been used to improve the resistance in example:

  • Line 32 ‘’Understanding the regulatory mechanisms that govern plant responses to multiple stresses comprises one of the most important features of biotechnological agriculture…’’’, here some example of practice usage will greatly enrich the manuscript. On the other hand, the plant breeding programmes and improving resistance to abiotic stresses by this should not be omitted here as well.

  • Attending your suggestion, we have added some paragraphs in the introduction discussing how this knowledge, supported by modern biotechnology, phenotyping techniques and genetic engineering can improve crop design and breeding. Please, check the section 1 – lines 72 – 114.

  • We have also discussed something close in the section reporting Programmed Cell Death Mechanisms – please, check the lines 579 – 593.

  • Differences between plant adaptation and acclimatization should be defined, explained, highlights and both terms should be used in proper meaning. Adaptation is mentioned in title, but there is no reference to adaptation in manuscript, please add it.

Since the paper focuses on plant responses and stress markers, overall, we have changed the title removing the term adaption, once we did not explore it.

  • There is a lack of secondary metabolism description as one of the important element of molecular response of plants to stress.

We have added a section of secondary metabolites as part of plant stress responses – please, check the section 4.

  • Table 1 should be signed and properly formatted, appropriate references of usage should be added.

Table 1 was completely revised and properly referenced.

Reviewer 5 Report

The authors attempted to review recent understanding of main mechanisms involved in plant response to abiotc stresses. Generally, this subject, being important both scientifically and for control of plant biomass production and thus relevant for global food security, is extensively explored and several reviews have been published around the issues addressed in the manuscript. The authors focused particularly on stress signals reception, the signals' transduction, particularly involvement of ROS, fitohormons and  other molecules as well as on the stress-induced senescence

This manuscript shows several unquestionable merits. It is well constructed and written very clearly, reflecting impressive scientific competence of the authors. Its advantage is also focusing on the practical markes of the stresses and 
gathered in extended Table, that could be very useful for potential reader.  The authors excellently combine various functional and structural aspect of plant response from ecological, agronomic to molecular level.
Moreover, the manuscript is almost ready for publication, in my opinion,  and I hope will find broad interest among researches involved in related subjects. 

Author Response

Dear reviewer,

We sincerely appreciate your positive comments over our manuscript and authors deeply acknowledge you by recognizing their competences and the potential of the manuscript. Attending the suggestions of other reviewers, we have added some information on how biotechnology, genetic engineering and breeding, supporting by the knowledge on plants stress responses and plant phenotyping, can improve the releasing of superior crops. Additionally, we have revised our Table of stress markers, referencing it properly and adding some information.